# Comparison of Regional Simulation of Biospheric CO$_2$ Flux from the Updated Version of CarbonTracker Asia with FLUXCOM and Other Inversions over Asia

**Samuel Takele Kenea \*, Lev D. Labzovskii, Tae-Young Goo, Shanlan Li, Young-Suk Oh and Young-Hwa Byun**

Climate Research Division, National Institute of Meteorological Sciences, 33, Seohobuk-ro, Seogwipo-si, Jeju-do 63568, Korea; labzovskii@korea.kr (L.D.L.); gooty@korea.kr (T.-Y.G.); sunranlee@korea.kr (S.L.); ysoh306@korea.kr (Y.-S.O.); yhbyun@korea.kr (Y.-H.B.)
\* Correspondence: samueltake@yahoo.ca; Tel.: +82-10-2328-0105

**Abstract:** There are still large uncertainties in the estimates of net ecosystem exchange of CO$_2$ (NEE) with atmosphere in Asia, particularly in the boreal and eastern part of temperate Asia. To understand these uncertainties, we assessed the CarbonTracker Asia (CTA2017) estimates of the spatial and temporal distributions of NEE through a comparison with FLUXCOM and the global inversion models from the Copernicus Atmospheric Monitoring Service (CAMS), Monitoring Atmospheric Composition and Climate (MACC), and Jena CarboScope in Asia, as well as examining the impact of the nesting approach on the optimized NEE flux during the 2001–2013 period. The long-term mean carbon uptake is reduced in Asia, which is −0.32 ± 0.22 PgC yr$^{-1}$, whereas −0.58 ± 0.26 PgC yr$^{-1}$ is shown from CT2017 (CarbonTracker global). The domain aggregated mean carbon uptake from CTA2017 is found to be lower by 23.8%, 44.8%, and 60.5% than CAMS, MACC, and Jena CarboScope, respectively. For example, both CTA2017 and CT2017 models captured the interannual variability (IAV) of the NEE flux with a different magnitude and this leads to divergent annual aggregated results. Differences in the estimated interannual variability of NEE in response to El Niño–Southern Oscillation (ENSO) may result from differences in the transport model resolutions. These inverse models' results have a substantial difference compared to FLUXCOM, which was found to be −5.54 PgC yr$^{-1}$. On the one hand, we showed that the large NEE discrepancies between both inversion models and FLUXCOM stem mostly from the tropical forests. On the other hand, CTA2017 exhibits a slightly better correlation with FLUXCOM over grass/shrub, fields/woods/savanna, and mixed forest than CT2017. The land cover inconsistency between CTA2017 and FLUXCOM is therefore one driver of the discrepancy in the NEE estimates. The diurnal averaged NEE flux between CTA2017 and FLUXCOM exhibits better agreement during the carbon uptake period than the carbon release period. Both CTA2017 and CT2017 revealed that the overall spatial patterns of the carbon sink and source are similar, but the magnitude varied with seasons and ecosystem types, which is mainly attributed to differences in the transport model resolutions. Our findings indicate that substantial inconsistencies in the inversions and FLUXCOM mainly emerge during the carbon uptake period and over tropical forests. The main problems are underrepresentation of FLUXCOM NEE estimates by limited eddy covariance flux measurements, the role of CO$_2$ emissions from land use change not accounted for by FLUXCOM, sparseness of surface observations of CO$_2$ concentrations used by the assimilation systems, and land cover inconsistency. This suggested that further scrutiny on the FLUXCOM and inverse estimates is most likely required. Such efforts will reduce inconsistencies across various NEE estimates over Asia, thus mitigating ecosystem-driven errors that propagate the global carbon budget. Moreover, this work also recommends further investigation on how the changes/updates made in CarbonTracker affect the interannual variability of the aggregate and spatial pattern of NEE flux in response to the ENSO effect over the region of interest.

**Keywords:** $CO_2$; NEE flux; CarbonTracker; FLUXCOM; Asia

---

## 1. Introduction

Due to the presence of strong anthropogenic emissions from fossil fuel consumption and cement production compared to the net uptake by the land and ocean, $CO_2$ accumulates in the atmosphere [1]. The ocean and biosphere are estimated to take up to $2.4 \pm 0.5$ and $3.0 \pm 0.8$ PgC $yr^{-1}$ of the $9.4 \pm 0.5$ PgC $yr^{-1}$ of anthropogenic carbon emitted to the atmosphere during the last decade [2]. Asia plays a key role in changing the global carbon cycle given its massive fossil fuel $CO_2$ emissions that are unlikely decrease in the near future [3], and its strong terrestrial carbon uptake. For example, Zhang et al. [4] found that the largest carbon sink in Asia was observed in forests, mainly in coniferous forests ($-0.64 \pm 0.70$ PgC $yr^{-1}$) and mixed forests ($-0.14 \pm 0.27$ PgC $yr^{-1}$); and the second and third largest carbon sinks were found in grass/shrublands and croplands, accounting for $-0.44 \pm 0.48$ and $-0.20 \pm 0.48$ PgC $yr^{-1}$, respectively. However, there are still large uncertainties in the quantified exchange of carbon between the atmosphere, ocean, and land reservoirs and these uncertainties are related to the terrestrial biosphere flux [2,5].

Asia is one of the regions with large uncertainties in the estimates of net ecosystem exchange of $CO_2$ (NEE). The uncertainties are driven by the sharp growth of fossil fuel emissions in most Asian countries. This growth has substantially affected the $CO_2$ balance of Asia and led to an increased variability of the regional carbon cycle [6,7]. Uncertainty in the fossil fuel emissions significantly contributes (32%) to the uncertainty in the land biosphere sink change as well [8]. The uncertainties are peculiarly large in the boreal and eastern part of temperate Asia, since the region has large land surface heterogeneity (i.e., land cover, vegetation growth, soil types, etc.). Also, climate change, land-use change, and human activities influence seasonal and interannual changes in NEE. All these factors hinder accurate estimation of NEE over Asia. Thus, the lack of a robust and precise quantification of the natural $CO_2$ fluxes in Asia limits our understanding about the links between the NEE flux and external forcing, such as meteorological variability, including the impact of extreme events, like droughts and cold spells, and trends in land-use change. Subsequently, forecasting the evolution of the land sink in Asia is also severely hindered. Moreover, all these factors affect the accuracy of the global budget of NEE estimates.

The inverse modeling estimates of $CO_2$ fluxes need an accurate simulation of observed atmospheric $CO_2$ concentrations since systematic model biases tend to directly translate into biased flux estimates. There have been many regional inverse modeling studies (e.g., Europe, North America) and many reported discrepancies in flux estimates and in their spatial distributions [9–13]. The discrepancies are mainly attributed to differences in modeling approaches (atmospheric transports, optimization methods, etc.), assumptions (e.g., different prior fluxes and uncertainties, domain definitions, etc.), and the accuracy of observations. Due to sparse coverage by observations and the poorly known error covariance structure, the NEE is to a large extent determined by the patterns of gross primary productivity (GPP) and terrestrial ecosystem respiration (TER) predicted by the original biosphere model. Locatelli et al. [14] pointed out that the impact of transport model uncertainty is varied in space and time. The other important aspect is that, in most inversions, the land cover classes do not fully encompass the ecosystem variability that results in the uncertainty of the flux estimates. Recently, Kenea et al. [15] assessed the accuracy of the CarbonTracker Asia version 2016 (CTA2016) model simulation of $CO_2$ concentrations using in-situ surface observations in East Asia and found that a better agreement is observed in most places during daytime than nighttime. Biases can emerge in the retrieved posterior $CO_2$, resulting from errors in the estimated fluxes or specific biases in transport to the location of the independent data or the representation of the planetary boundary layer (PBL).

We present the CarbonTracker Asia (CTA2017)-simulated NEE flux run by the National Institute of Meteorological Sciences (NIMS), Republic of Korea, along with the global inversion models CT2015

to CT2017 (documented at http://carbontracker.noaa.gov) [16], Copernicus Atmospheric Monitoring Services (CAMS) [17,18] (http://apps.ecmwf.int/datasets/data/macc-ghg-inversions/), Monitoring Atmospheric Composition and Climate (MACC-III) [17] (http://apps.ecmwf.int/datasets/data/cams-ghg-inversions/), Jena CarboScope inversion system developed at the MPI for Biogeochemistry Jena [19,20], and FLUXCOM (https://doi.org/10.17871/BACI.224) [21]. Atmospheric transport inversion depends on transport models, statistical methodologies, a priori information, number of observation data used for assimilation, and model resolution to derive the most likely estimates of NEE fluxes. The following updates were made in CTA2017: The assimilation window was extended to 12 weeks while in the previous version, the assimilation window was chosen to be 5 weeks long; the vertical diffusion scheme was changed to the Yonsei University planetary boundary layer (YSU PBL) scheme; land biosphere prior uncertainty was larger than the previous release; and the horizontal transport resolution was $1° \times 1°$ degrees of latitude and longitude while in CT2017, the resolution is $2° \times 3°$ degrees of latitude and longitude. Therefore, the analysis focuses on analyzing and comparing CTA2017-simulated NEE flux with FLUXCOM and other global inversions in terms of spatial and temporal distribution over Asia, as well as over individual biomes. In short, we address the following questions in this work:

- What is the consistency and discrepancy of NEE flux of CTA2017 compared to MACC, CAMS, Jena CarbonScope, and FLUXCOM in terms of monthly, seasonal, and annual time scales?
- How consistent are the CTA2017 estimates of spatial diurnal averaged NEE flux in comparison with the FLUXCOM?
- What is the impact of the nesting approach on the strength of the carbon sink and source over Asia at seasonal and annual timescales?

## 2. Materials and Methods

### 2.1. CarbonTracker CO₂ Flux

The CarbonTracker model is a carbon data assimilation system for $CO_2$ estimates of global carbon and sources and sinks and was originally developed by the Earth System Research Laboratory (ESRL) at the National Oceanic and Atmospheric Administration Earth System (NOAA). Subsequently, development was carried out on CarbonTracker (NOAA/ESRL), and CarbonTracker Asia [16,22]. Here, the focus is on the CarbonTracker Asia system.

The CarbonTracker data assimilation system for $CO_2$ estimates the carbon fluxes between the atmosphere, land biosphere, and oceans, using atmospheric observations of $CO_2$ mole fractions. The total $CO_2$ fluxes, $F(x, y, t)$, for each region, $r$, defined by longitude ($x$) and latitude ($y$) and each time step ($t$) are represented by Equation (1):

$$F(x, y, t) = \lambda_r F_{bio}(x, y, t) + \lambda_r F_{ocn}(x, y, t) + F_{ff}(x, y, t) + F_{fire}(x, y, t), \qquad (1)$$

where $F_{bio}$, $F_{ocn}$, $F_{fire}$, and $F_{ff}$ are prior flux model predictions for land biosphere, ocean, wildfire, and fossil fuel emission, respectively, and $\lambda_r$ is a scaling factor estimated for each week and is assumed to be constant over this period for a particular region, rather than attempting to adjust fluxes within individual grid cells to reduce the dimensions of the inversion problem within the CarbonTracker system. To adjust the carbon fluxes via optimization of the scaling factor, observed atmospheric $CO_2$ concentrations were assimilated with an atmospheric transport TM5 offline model [23] using an ensemble Kalman filter methodology. The selection of regions that has a strong a priori constraint on the derived fluxes should be thoughtfully treated to avoid so-called "aggregation errors" [24]. The model embraces a nested-grid approach to optimize the surface $CO_2$ fluxes over the region of interest. The outer boundaries of the grid were driven by the global scale simulation. TM5 was driven by ERA-Interim assimilated meteorology from European Centre for Medium-Range Weather Forecasts (ECMWF) at the spatial resolution of 1° by 1° degrees [25]. The increased spatial resolution could change the vertical mixing of the species because deep convective mixing is a sub-grid scale process.

In this work, we used the CTA2017 model NEE flux that was run by the National Institute of Meteorological Sciences (NIMS), Korea. Similar to the earlier version of CarbonTracker, the same biospheric prior fluxes from two variants of the Carnegie-Ames-Stanford Approach (CASA) models were used: Global Fire Emissions Database (GFED 4.1s) (at the spatial resolution of 0.25°) and GFED_CMS (at the spatial resolution of 0.5°) at the temporal resolution of 90 min. The following changes were made in CTA2017: The assimilation window was extended to 12 weeks in order to enhance resolving fluxes in regions with less dense observational coverage while in the previous version, the assimilation window was chosen as 5 weeks long. In CTA2017, the vertical diffusion scheme was changed to the YSU PBL scheme, which differs from CT2017, since YSU allows greater vertical mixing, diffusing $CO_2$ in the PBL instead of trapping it in the surface layer. The horizontal resolution of the transport model was $1° \times 1°$ degrees of latitude and longitude in CTA2017 while in CT2017, the resolution was $2° \times 3°$ degrees of latitude and longitude. The land biosphere prior uncertainty was larger in CTA2017 compared to the previous release [16,22]. The accuracy of simulated NEE can be improved by assimilating the observation data. For optimization of the NEE flux, the surface in-situ and flask sites data obtained from GLOBALVIEWplus-3.1 ObsPack (https://www.noaa.gov/gmd/ccgg/carbontracker/) and Comprehensive Observation Network for Trace gases by AIrLiner (CONTRAIL) $CO_2$ concentration data were used. Note that in-situ site data that are close to the polluted area or within the nocturnal boundary layer were excluded in order to avoid the potential biases in flux estimations. In CTA2017, the assimilated sites that differ from the global simulation of CT2017 are Ryori, Yonagunijima, and Minamitorishima and also aircraft observation data from CONTRAIL. The use of data from remote sites, such as Yonagunijima and Minamitorishima, in the assimilation system is not as impactful in affecting the land NEE flux. The way we used the CONTRAIL aircraft observation data in CarbonTracker Asia is similar to the approach suggested by Zhang et al. [4]. The optimized NEE flux in 2000 was excluded from this analysis since this period was considered a spin-up year.

## 2.2. FLUXCOM Data

We used gridded NEE flux data (https://doi.org/10.17871/BACI.224) [21] over Asia from observation-driven global half-hourly and monthly resolution data from the FLUXNET measurements obtained from the Portal of the Max Planck Institute for Biochemistry (https://www.bgc-jena.mpg.de). The product has a spatial resolution of $0.5° \times 0.5°$ and covers the period from 1982 to 2013. To be consistent with CTA2017, we used the data from the 2001 to 2013 periods. FLUXNET is a large-scale measurement network consisting of 224 sites (Figure 1) that are equipped with eddy covariance towers, and provides carbon fluxes across biomes and climates globally [26]. However, eddy covariance measurements were site-level observations (at <1 km$^2$ scale), so that spatial upscaling was applied to estimate NEE fluxes at regional to global scales. The statistical upscaling techniques employed machine learning methods, such as random forests (RFs), multivariate regression splines (MARS), and artificial neural network (ANN) ([26] and references therein), utilizing satellite remote sensing, climate and meteorological data, and land cover information on the site observation fluxes [27].

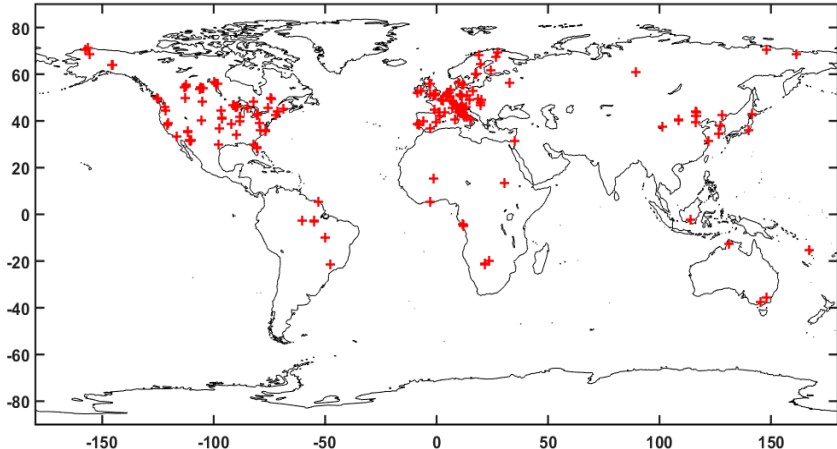

**Figure 1.** Eddy covariance flux tower sites used for FLUXCOM.

## 2.3. Other Atmospheric Inversion Data

In this work, we utilized three inversion NEE flux data, including Copernicus Atmospheric Monitoring Services (CAMS), Monitoring Atmospheric Composition and Climate (MACC-III), and the Jena CarboScope inversion system developed at the MPI for Biogeochemistry Jena [19,20]. The CAMS [17,18] inversion data (version v18r1) (http://apps.ecmwf.int/datasets/data/cams-ghg-inversions/) were provided for 1979–2018 with a monthly mean time step and a spatial resolution of 1.875° latitude and 3.75° longitude. The MACC-III [17] inversion data (version v14r1_ra) (http://apps.ecmwf.int/datasets/data/macc-ghg-inversions/) were provided for 1979–2014 with a monthly mean time step and a spatial resolution of 1.875° latitude and 3.75° longitude. MACC-III $CO_2$ flux inversion used NOAA Carbon Cycle Greenhouse Gases (CCGG), World Data Centre for Greenhouse Gases (WDCGG), and RAMECES observation networks for assimilation. The Jena CarboScope inversion (s04_v4.2) provides daily fluxes at a spatial resolution of 3.75° latitude and 5° longitude. Fluxes from Jena CarbonScope (Jena inversion) are optimized at the spatial resolution of the transport model from weekly to interannual time scales. The model employed the variational approach to assimilate observations of $CO_2$ with the transport model. Table 1 provides information about the features of the inverse models.

**Table 1.** List of the inverse modeling systems used in this work. Note that inversions account for interannually varying (IAV) prior fluxes (Yes or No) and transport (Yes or No).

| Model | Ref. | Lat × lon | Fossil Fuel Priors | Biosphere and Fire Priors | Ocean Priors | Transport Model | No Vertical Layers | Meteorological Fields | Time Span | IAV Priors | IAV Wind |
|---|---|---|---|---|---|---|---|---|---|---|---|
| CT2017 | [15] | 2° × 3° global 1° × 1° over Asia | ODIAC2016 and Miller | CASA (GFED 4.1s & GFED_CMS) | Jacobson et al. [28] and Takahashi et al. [29] | TM5 | 25 | ERA-Interim | 2001–2013 | Yes | Yes |
| Jena (s04_v22) | [19] | 3.8298° × 5° | CDIAC | | | TM3 | 19 | NCEP | 2004–2013 | No | Yes |
| CAMS (v18r1) | [16,17] | 1.8947° × 3.75° | CDIAC/GCP2016 | ORCHIDEE (climatology) +GFEDv4 | Takahashi et al. [29] | LMDZ | 39 | ERA-Interim | 2001–2013 | Yes | Yes |
| MACC (v14r2) | [16,17] | 1.8947° × 3.75° | CDIAC/GCP2016 | ORCHIDEE (climatology) +GFEDv4 | Takahashi et al. [29] | LMDZ | 39 | ERA-Interim | 2001–2013 | Yes | Yes |

## 2.4. Land Cover Data

To assess the impact of the land cover map on carbon flux, we used Moderate Resolution Imaging Spectroradiometer (MODIS) land cover data (MCD12Q1 version 051 of the year 2013) along with a map of [30] used by CTA2017 (note that other versions of CarbonTracker also used the same assumption of land cover mapping). The MODIS land cover map was re-sampled into a 1° × 1° degree spatial resolution that corresponds to CTA2017 model (all versions). Figure 2 depicts the

distribution of ecosystem types over Asia taken from CTA2017 (top panel) and MODIS (bottom panel). We noted a mismatch between them, for example, CAT2017 depicted a forest field land within the cropland in northeast China, but this land type was not shown by MODIS. Such disparity of ecosystem representations is attributed to the discrepancy between the inverted fluxes and upscaled carbon fluxes. There has been significant changes in land cover in Asia over the period between 1990 and 2010; the forest extent increased in East Asia by ~22% (45.4 Mha), and South Asia by ~3% (2.1 Mha), whereas Southeast Asia experienced a major loss of forest extent, which is estimated to be ~13% (33.2 Mha) [7]. Stibig et al. [31] also indicated that the total forest cover of Southeast Asia was estimated at 268 Mha in 1990, dropping to 236 Mha in 2010, with annual change rates of 1.75 (~0.67%) and 1.45 Mha (~0.59%) for the periods 1990–2000 and 2000–2010, respectively. East Asia ecosystems are a substantial sink of $CO_2$, taking up an estimated 0.16 to 0.33 PgC $yr^{-1}$ due to afforestation/reforestation and regional climate change, especially in southern China ([8] and references therein).

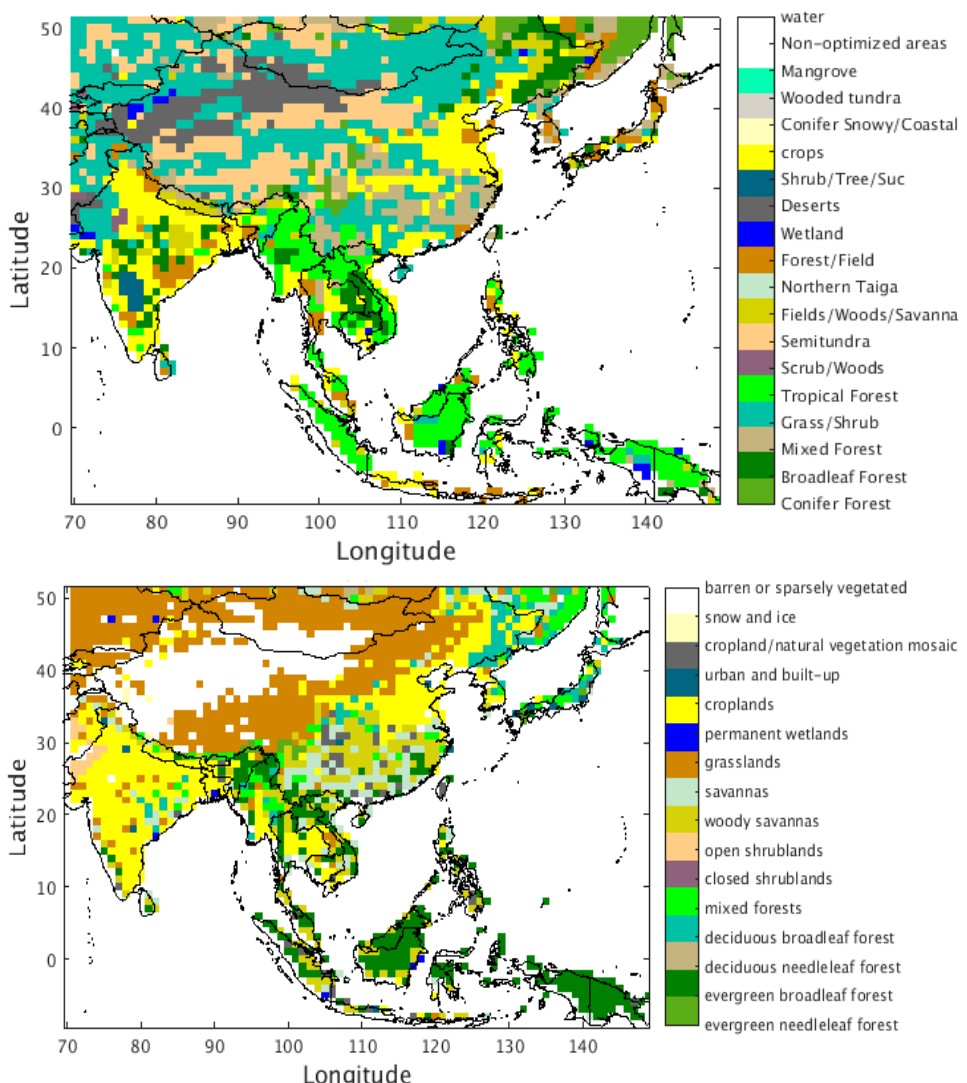

**Figure 2.** Land ecosystems classification used by the CarbonTracker Asia version 2017 CTA2017 (all versions used the same assumption) (**top panel**), and the Moderate Resolution Imaging Spectroradiometer (MODIS) observations 2013 (**bottom panel**).

*2.5. Methods*

We presented the spatio-temporal distribution of NEE flux over Asia (−9.5° N–51.5° N latitude and 69.5°E–149.5° E longitude). The comparison of annual and summer NEE spatial distributions

and the corresponding spatial correlations between CTA2017 and FLUXCOM is presented during 2001–2013. All annual, summer, and correlation estimates are based on monthly mean NEE estimates of the individual grid cell. Note that we applied the spatial re-gridding on the FLUXCOM data from 0.5° by 0.5° to 1.0° by 1.0° in order to match the CarbonTracker model resolution. We used Pearson's correlation coefficient to see the strength of the association among the target parameters. The spatial Pearson's correlation coefficient of the monthly mean NEE values between CTA2017 and FLUXCOM taken over each grid cell during 2001–2013 was obtained using Equation (2):

$$r_{xy} = \frac{\sum_i^n (x_i - \overline{x})(y_i - \overline{y})}{\sqrt{\sum_i^n (x_i - \overline{x})^2} \ \sqrt{\sum_i^n (y_i - \overline{y})^2}}, \tag{2}$$

where $n$ is the total number of data points, $x_i$ is CTA2017 NEE, $y_i$ is FLUXCOM NEE, and $\overline{x}$ and $\overline{y}$ are the mean values for each grid cell. In order to see the level of agreement (between CTA2017 with FLUXCOM, CAMS, MACC, Jena CarbonScope, and CarbonTracker global versions) in terms of the time series, we showed NEE flux integrated over the domain and globally on the yearly and annual mean (PgC yr$^{-1}$). Besides, the temporal evolution of monthly NEE (gCm$^{-2}$ yr$^{-1}$) averaged over individual biomes (conifer forest, broadleaf forest, mixed forest, grass/shrub, tropical forest, shrub/woods, fields/woods/savanna, and cropland) was also discussed to examine the consistency between CTA2017 and FLUXCOM. To further explore the agreement of the diurnal cycle of NEE between CTA2017 and FLUXCOM, we provided 3 hourly NEE averages (3-hour intervals) for the whole year (2001–2013) for each grid cell of a given domain from June to August. To examine the impact of the nesting on the strength of the carbon sink and source, we compared the spatial and temporal aggregation of NEE fluxes between CTA2017 and CT2017 in terms of annual and seasonal time scales.

## 3. Results and Discussion

### 3.1. Assessment of Consistency and Discrepancy of Spatial and Domain Aggregated NEE Flux

Here, we compared the spatial distributions of annual and summer means of NEE of CTA2017 with FLUXCOM estimates (Figures 3 and 4). Note that the area where the FLUXCOM data show a large number of missing points is over the bare land. The overall pattern of the annual mean NEE indicates that the inversion model underestimated the carbon uptake against FLUXCOM across the entire region of the domain. The NEE values from CTA2017 varied within −0.5 to 0.5 gC m$^{-2}$ d$^{-1}$ while values from FLUXCOM range from −2.5 to 0 gC m$^{-2}$ d$^{-1}$, with the maximum carbon sink observed over the southeast part of the region. Furthermore, a spatial Pearson's correlation coefficient was computed between CTA2017 and FLUXCOM based on the monthly mean values. The result offers a maximum negative correlation of −0.63, over Southeast Asia that is mainly comprised of Nepal, Bangladesh, Bhutan, and Myanmar, as shown in Figure 3d. As seen from the NEE pattern in summer between CTA2017 (prior (a) and posterior (b)) and FLUXCOM (c) in Figure 4, the result implied that the inversion exhibited a prevalent carbon source in south and Southeast Asia, whereas the FLUXCOM showed a contrasting pattern in this part of the region. In this sub-region, the posterior flux from the inversion is strongly influenced by the prior flux, since the posterior NEE estimation has a small departure from the prior estimation (Figure 4a,b). The land cover inconsistency is one of the main drivers for the discrepancy of the NEE estimates. We discerned some mismatches on individual pixel-wise biomes that were assumed by CTA2017 and MODIS observations in a region where a strong negative correlation obtained. A negative correlation shown in Japan and northeastern China was mostly confined in forest fields, which are based on the land cover classification category used by the inversion. According to the MODIS land cover classification, those forest field covers were not shown in those regions; rather, it depicted mixed forest, cropland, and grassland. Earlier inversion studies pointed out that the optimized fluxes were sensitive to prior fluxes, particularly for the regions that

were poorly constrained by atmospheric observations, such as the tropics [32]. The studies indicated in, for example, [6,33,34] that the inverted fluxes were sensitive to observation data. In addition to that, the challenges in data-driven estimation of $CO_2$ fluxes in tropical forests were also described by Tramontana et al. [26]. The challenges emerge in extreme environments for reduced water resources (dry) or low temperature (cold), and in managed sites, such as croplands (crop). Ichii et al. [35] also found inconsistency in the spatial estimated NEE between the data-driven and Greenhouse Gases Observing Satellite (GOSAT) level 4A over south Asia and Southeast Asia, which was partially clarified by accounting for the differences in the definition of $CO_2$ fluxes. The area where large discrepancy occurred between the inversion and the FLUXCOM needs further work (e.g., increasing assimilated data from space-borne, improving priori covariance in the inversion, and improving the upscaling approach to improve $CO_2$ budget estimates).

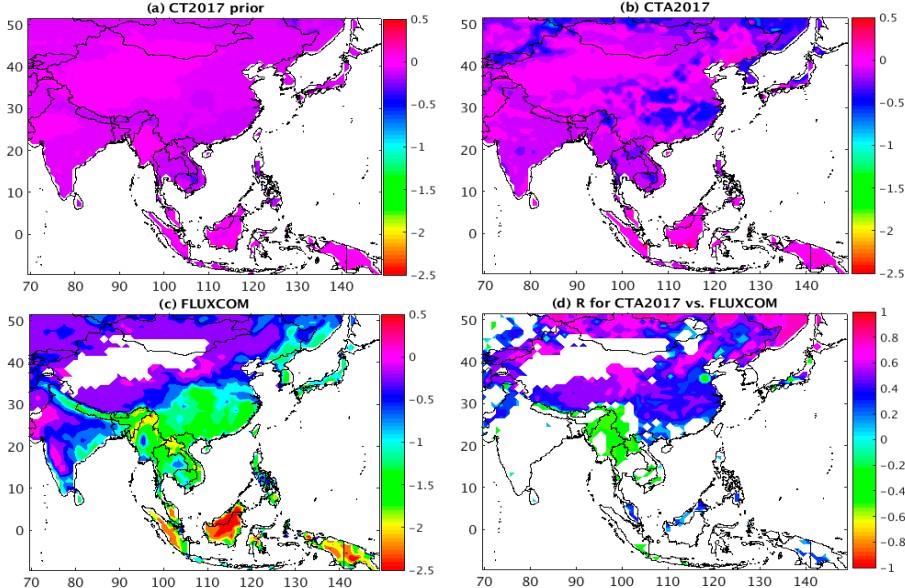

**Figure 3.** Annual net ecosystem exchange of $CO_2$ (NEE) flux spatial distribution (gC m$^{-2}$ d$^{-1}$) from CT2017 prior (**a**), CTA2017 posterior (**b**), FLUXCOM (**c**), and correlation between CTA2017 and FLUXCOM (**d**) for the 2001–2013 period. Pearson's correlation coefficient (R) values shown are significant at a 95% confidence interval.

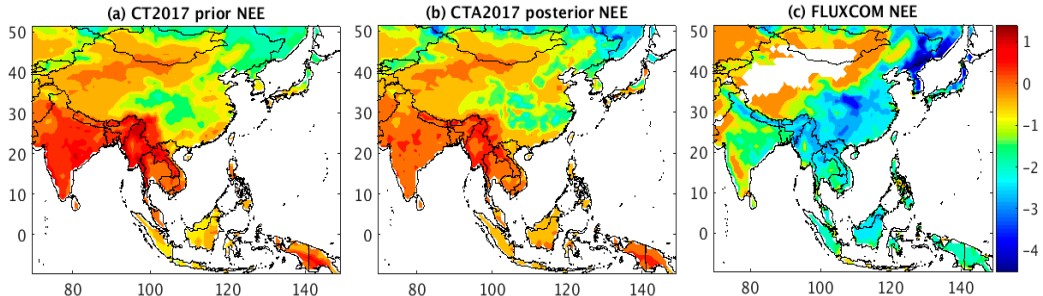

**Figure 4.** Summer NEE flux spatial distribution (gC m$^{-2}$ d$^{-1}$) from CT2017 prior (**a**), CTA2017 posterior (**b**), and FLUXCOM (**c**) for the 2001–2013 period.

In this section, we presented the time series of yearly posterior NEE flux (natural + fires) integrated over Asia from several inversion systems: CT2015, CT2016, CT2017, CTA2016, CTA2017, JENA, CAMS, and MACC along with the FLUXCOM. Since the inversion fluxes obviously have large uncertainties at the pixel level, we focused on the analysis of the comparison based on the aggregated NEE flux from Asia and the global scale. Figure 5 depicts the time series of yearly aggregated NEE flux over the global scale (a) and Asia (b). Globally, all models revealed an increasing trend of the carbon sink between

2001 and 2011 and a decreasing trend from 2011 onwards (Figure 5a). In addition, the interannual variability of the land carbon sink is evident in all models, but the magnitude is different. Note that the inverted interannual variability of (IAV) NEE flux presented here was constrained by observations, priors, and wind. The land carbon sink was the lowest in 2002 and was the highest in 2011. The weaker land carbon sinks in 2002, 2005, 2007, and 2010 correspond to strong El Niño events while the stronger land sink in 2008 and 2011 corresponds to strong La Niña events. The IAV of the land carbon sink is mainly driven by the ENSO, which is dominated by tropical land fluxes [36]. Globally, in each individual year, the estimates of carbon sink from all versions of CarbonTracker are smaller than CAMS and MACC, which might be mostly explained by the use of different prior flux. As seen from Figure 5b, such an IAV of the yearly aggregated NEE estimates from Asia is also explicitly observed by the models. For example, both CTA2017 and CT2017 captured the IAV, but their magnitudes are different, and this leads to divergence of the annual aggregated NEE results.

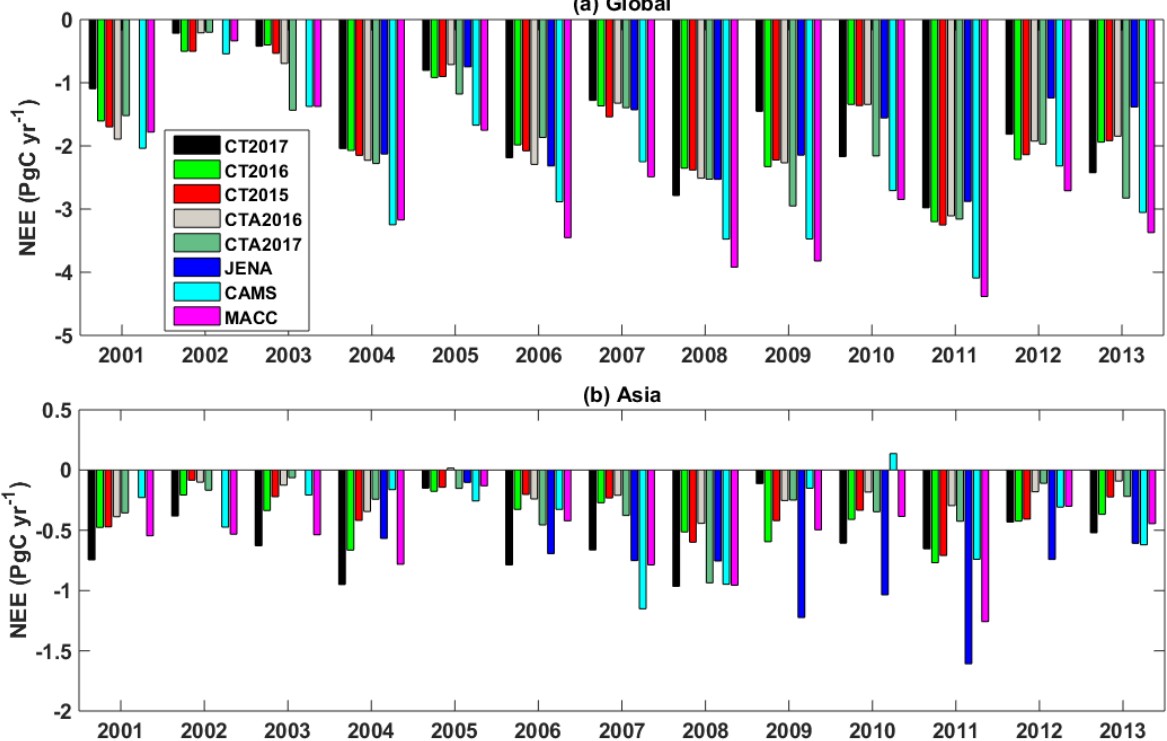

**Figure 5.** The time series of yearly posterior NEE flux integrated over the global scale (**a**) and Asia (**b**) derived from CT2017, CT2016, CT2015, CTA2016, CTA2017, JENA, Copernicus Atmospheric Monitoring Services (CAMS), Monitoring Atmospheric Composition and Climate (MACC) inversions.

At our domain scale, CTA2017 estimated the annual average aggregated carbon sink to be $-0.32$ PgC yr$^{-1}$, which is smaller than CT2017, CAMS, MACC, Jena, and FLUXCOM. The annual average of CT2017, CAMS, MACC, Jena, and FLUXCOM estimates were $-0.58$, $-0.42$, $-0.58$, $-0.81$, and $-5.86$ PgC yr$^{-1}$, respectively (Table 2). A difference with FLUXCOM was mainly due to differences in the forest in the tropics. Errors in the transport model, inverse model set-up, and lack of data for assimilation are contributors for the disagreement in estimates of NEE among the inversion models. For example, the resulting discrepancy between MACC and CAMS might be driven by the differences in the number of data used for assimilation, since they have the same transport model and inverse set-up. As we have seen for the IAV, quantified as the standard deviation of annual NEE, the CT2017 are higher than the earlier version of the CarbonTracker, and less than CAMS, MACC, and Jena CarboScope by 63.6%, 36.4%, and 86.4%, respectively. In each year during the 2001–2013 period, the aggregated sum of the carbon sink flux obtained from CTA2017 was lower than CT2017. Similarly, this pattern was also observed in CT2016. This pattern most likely arises from a higher resolution in atmospheric

transport. There is also an increase of the carbon uptake in Asia as we see the series of model versions from CT2015 to CT2017 by 26.5% (CT2016) and 70.6% (CT2017) with respect to CT2015. Very recently, Hu et al. [37] discussed the difference in the North American NEE response to ENSO from different resolutions of CarbonTracker models, which used a similar suite of atmospheric $CO_2$ observations and the same prior fluxes. They pointed out that the difference in the estimated NEE response to ENSO may result from the difference in the transport models' resolutions.

**Table 2.** Annual NEE flux (natural + fire flux) integrated over the global scale and Asia for 2001–2013, except for Jena CarboScope (2004–2013), is provided. Unit is given in PgC yr$^{-1}$.

| Models/ Upscale Data | NEE Flux (PgC yr$^{-1}$) | | |
|:---:|:---:|:---:|:---:|
| | Asia Posterior | Prior | Global Posterior |
| CTA2017 | −0.32 ± 0.22 | 0.25 ± 0.13 | −1.96 ± 0.82 |
| CTA2016 | −0.22 ± 0.13 | 0.25 ± 0.13 | −1.72 ± 0.83 |
| CT2017 | −0.58 ± 0.26 | | −1.71 ± 0.80 |
| CT2016 | −0.43 ± 0.18 | | −1.71 ± 0.80 |
| CT2015 | −0.34 ± 0.18 | | −1.74 ± 0.78 |
| CT2013B | 1.01 ± 1.50 | | −2.36 ± 4.84 |
| CAMS | −0.42 ± 0.36 | −0.08 ± 0.007 | −2.55 ± 0.98 |
| MACC-III | −0.58 ± 0.30 | −0.08 ± 0.007 | −2.72 ± 1.15 |
| Jena CarboScope | −0.81 ± 0.41 | | −1.84 ± 0.66 |
| FLUXCOM | −5.86 ± 0.03 | | |
| Thomson et al. [8] | −0.46 (Asia, 1996–2012) | | |
| Ichii et al. [35] | −0.90 (East Asia) −0.94 (Southeast Asia) | | |

We checked the consistency and discrepancy of the simulated monthly mean fossil fuel $CO_2$ (FFCO$_2$) flux from CTA2017 and CT2017. Figure 6 depicted the time series of monthly FFCO$_2$ flux averaged over the domain for 2001–2013. They imparted somewhat large differences on FFCO$_2$ flux at monthly time scales. Notably, we observed the sinusoidal patterns, with a larger amplitude seen by the nested grid simulation. In 2001–2007, all peaks in CTA2017 were higher than in CT2017, but the minimum values were almost similar. In the period between 2001 and 2009, the change of simulated FFCO$_2$ flux with and without nesting experiments yielded 71.6% and 78.0%, respectively. Bear in mind that all FFCO$_2$ settings were similar in both simulations except for the horizontal resolution difference in the TM5. In a similar period, the overall aggregated NEE magnitude appeared to be higher (i.e., lower carbon uptake) in CTA2017 (Figure 5b). Some artifact can be induced on retrieved NEE flux due to the uncertainty of the fossil fuel estimations [9,38]. Saeki and Patra [38] demonstrated that the $CO_2$ sink increase estimated from the inversion model over Asia was about 0.26 PgC yr$^{-1}$ during 2001–2010. This was likely an artifact of the anthropogenic $CO_2$ emissions increasing too quickly (by 1.41 PgC yr$^{-1}$) in China. The magnitude of this increase in East Asia is contingent on the assumed increase in FFCO$_2$ emissions (inventories differ by up to 17% in East Asia).

We also assessed the level of agreement between CTA2017 and FLUXCOM in producing the temporal variation of domain aggregated NEE flux (Figure 7). Even though the overall results demonstrated that both captured the seasonal cycle of NEE, they exhibited a clear systematic difference. For 2001–2013, CTA2017 averaged values were between −0.71 and 0.28 gC m$^{-2}$ d$^{-1}$ for much of the year while the FLUXCOM varied from −0.20 to −1.53 gC m$^{-2}$ d$^{-1}$. Their minimum and maximum differences were 71.6% and 171.3% in winter and summer, respectively. In addition, an interannual variation was barely seen in the FLUXCOM. On the interannual timescale, variations in biospheric uptake can be partially attributed to climate, phenology, physiology, and natural and anthropogenic disturbances [39,40].

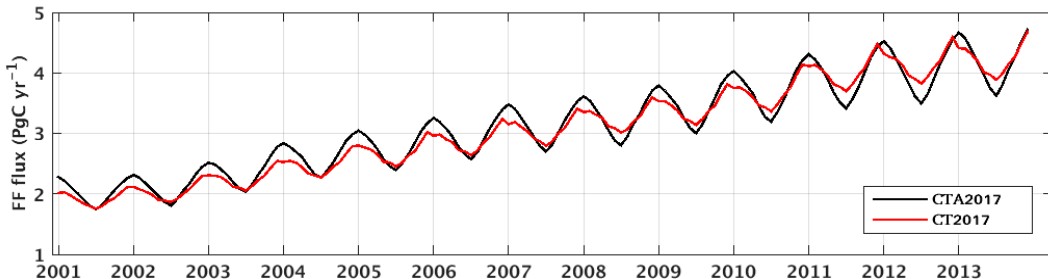

**Figure 6.** Time series of domain (Asia) aggregated monthly mean fossil fuel flux from CTA2017 and CT2017.

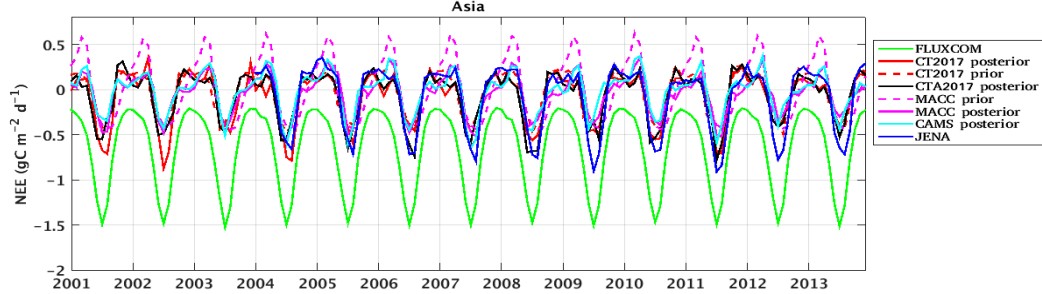

**Figure 7.** The time series of monthly domain averaged NEE flux derived from FLUXCOM, CTA2017, CT2017, MACC, CAMS, and Jena.

The time series of the monthly domain averaged NEE from CAMS, MACC, and Jena CarboScope was also compared with FLUXCOM during 2001–2013 (Figure 7 and Table 3). The result is similar to the comparison results of CTA2017 versus FLUXCOM, exhibiting a large systematic difference. As stated in the previous section, such a difference is mainly attributed to the large carbon uptake estimates from FLUXCOM in tropical Asia. We also obtained correlation coefficients of 0.83 for CAMS versus FLUXCOM, 0.80 for MACC versus FLUXCOM, and 0.94 for Jena versus FLUXCOM. Deeper analysis of the individual biomes is impeded by the coarser horizontal resolution of the inversion models compared to the FLXUCOM resolution. We also examined the winter and summer means of the priori and posterior NEE flux among the MACC, CAMS, CTA2017, and CT2017. In winter, the estimated prior and posterior NEE flux from MACC and CAMS is 0.26 gC m$^{-2}$ d$^{-1}$ and about 0.07 gC m$^{-2}$ d$^{-1}$, respectively, whereas CTA2017 and CT2017 exhibited 0.18 gC m$^{-2}$ d$^{-1}$ and about 0.11 gC m$^{-2}$ d$^{-1}$, respectively. In comparing the differences (posterior minus prior fluxes) of CTA2017 and CT2017 with CAMS and MACC, the magnitude of such a deviation is slightly larger in MACC and CAMS, thus suggesting the influence of prior MACC and CAMS is somehow less as compared to CTA2017 and CT2017 (Table 3). However, the reverse holds true during summer.

**Table 3.** Winter and summer means and the corresponding standard deviations of domain averaged NEE flux for 2001–2013, except for Jena CarboScope (2004–2013), is presented.

| Models/ Upscale Data | Prior (gCm$^{-2}$d$^{-1}$) | | Posterior (gC m$^{-2}$ d$^{-1}$) | |
|---|---|---|---|---|
| | Winter | Summer | Winter | Summer |
| CTA2017 | 0.18 ± 0.03 | −0.37 ± 0.08 | 0.11 ± 0.06 | −0.49 ± 0.13 |
| CT2017 | 0.18 ± 0.03 | −0.37 ± 0.08 | 0.10 ± 0.08 | −0.50 ± 0.16 |
| CAMS | 0.26 ± 0.10 | −0.37 ± 0.07 | 0.07 ± 0.04 | −0.36 ± 0.13 |
| MACC-III | 0.26 ± 0.10 | −0.37 ± 0.07 | 0.06 ± 0.06 | −0.36 ± 0.12 |
| Jena CarboScope | – | | 0.18 ± 0.07 | −0.64 ± 0.14 |
| FLUXCOM | | | −0.24 ± 0.03 | −1.34 ± 0.12 |

### 3.2. Comparisons of Temporal Distributions of Aggregated Mean NEE Flux across Biomes

In order to examine the ability of CTA2017 and CT2017 to simulate NEE across individual biomes, the seasonal and annual averaged NEE for 2001–2013 was compared for various biomes. This means that areas for model improvement can be identified on scales smaller than the region of interest. Here, we analyzed the time series of aggregated mean NEE flux across biomes in Asia between CTA2017, and CT2017 versus FLUXCOM.

Figure 8 provides the time series of monthly NEE over conifer forest, broadleaf forest, mixed forest, grass/shrub, tropical forest, shrub/woods, fields/woods/savanna, and croplands in Asia from CTA2017, CT2017, and FLUXCOM. The overall results revealed good consistency of the seasonal variations over conifer forest, broadleaf forest, mixed forest, grass/shrub, and fields/woods/savanna; poor agreement in croplands and shrub/woods; and the worst agreement in tropical forest land. The croplands and scrub/woods land of different climate zones might be the drivers of poor agreement for seasonal variations between the CarbonTracker models and FLUXCOM. We also indicated the Pearson's correlation coefficient of NEE of CTA2017 and CT2017 with FLUXCOM across biomes and the result suggested that both were well correlated in conifer forest (Table 4), with a correlation coefficient of 0.87. Besides, we obtained a linear regression slope of 0.82 (Figure 9b and Table 4). In mixed forest lands, all depicted a clear seasonal cycle of NEE, and further confirmed the level of agreement between CTA2017 and CT2017 with FLUXCOM, with correlation coefficients of 0.79 and 0.70, respectively. The inverse models exhibited an evidently decreasing carbon uptake, particularly for 2002–2007.

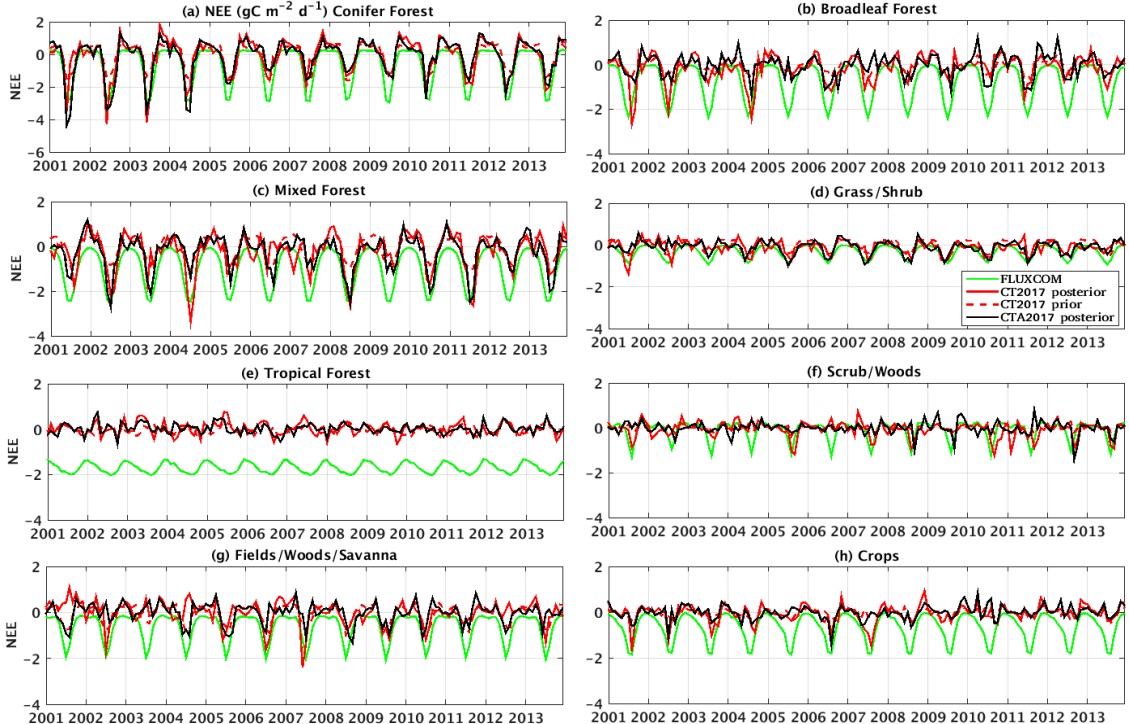

**Figure 8.** Time series of monthly NEE flux averaged over individual biomes in Asia during 2001–2013 is shown.

The annual mean NEE of broadleaf forest showed some differences between the estimates of CTA2017 (–0.04 ± 0.50 gC m$^{-2}$ d$^{-1}$) and CT2017 (–0.16 ± 0.60 gC m$^{-2}$ d$^{-1}$) than the rest of the ecosystem types (Figure 9 and Table 5), which is caused by the nesting effect. Both inverse estimates determined had comparable annual mean NEE flux over conifer forest, mixed forest, grass/shrub, and scrub/woods ecosystems. Notably, in winter, the inversions depicted large positive flux values with large variability, whereas the FLUXCOM was either close to zero or negative values with very small variability. For example, in conifer forest, the peak carbon uptake and source occurred in

summer (about $-4.20$ gC m$^{-2}$ d$^{-1}$ in 2001 and 2002) and in winter (around 2.0 gC m$^{-2}$ d$^{-1}$ in 2004), respectively, as observed by both inversions and FLUXCOM. Moreover, both CarbonTracker models portrayed a decreasing trend in carbon uptake for the period of 2001–2008, and an increasing trend for 2009–2013. On the one hand, a strong drawdown of carbon across all ecosystem types was exceptionally determined by the FLUXCOM, which was not observed by the inversion models (Figure 9a). It is difficult to reconcile FLUXCOM with the inversions. As can be seen in in Figure 8e, there was a significant discrepancy, on average of $-1.70$ gC m$^{-2}$ d$^{-1}$, on the time series of monthly NEE in tropical forest. The peak-to-trough amplitude of the monthly mean seasonal cycle was clearly demonstrated by the FLUXCOM but not in the inversion models. These disagreements could be attributed to the limits of the bottom–up approach in dealing with the low seasonality of the fraction of absorbed radiation (FaPAR) in evergreen broadleaf forests, and underrepresentation of NEE estimates by limited eddy covariance flux measurements. Alternatively, the discrepancies can be explained by the role of $CO_2$ emissions from land use change, which are particularly relevant in some tropical areas but were not accounted for in the FLUXCOM estimates [41].

**Table 4.** Summary of the statistical relationships of NEE flux averaged over individual biomes between CTA2017 and CT2017 with FLUXCOM in Asia. Note that all the correlation coefficients are statistically significant ($p < 0.05$) and * indicates insignificance. Pearson's correlation coefficients (r1) and slope (m1) represent for CTA2017 vs. FLUXCOM, while r2 and m2 for CT2017 vs. FLUXCOM. Unit for NEE flux is gC m$^{-2}$ d$^{-1}$.

| S.N | Land Cover Type | r1 | r2 | m1 | m2 |
|:---:|:---:|:---:|:---:|:---:|:---:|
| 1 | Conifer forest | 0.87 | 0.87 | 0.87 | 0.82 |
| 2 | Broadleaf forest | 0.67 | 0.68 | 0.42 | 0.50 |
| 3 | Mixed forest | 0.79 | 0.70 | 0.69 | 0.65 |
| 4 | Grass/shrub | 0.72 | 0.58 | 0.77 | 0.61 |
| 5 | Tropical forest | $-0.15$ | $-0.12$ * | $-0.15$ | $-0.14$ |
| 6 | Scrub/woods | 0.20 | 0.49 | 0.14 | 0.41 |
| 7 | Fields/woods/savanna | 0.60 | 0.52 | 0.45 | 0.45 |
| 8 | Croplands | 0.43 | 0.63 | 0.22 | 0.4 |

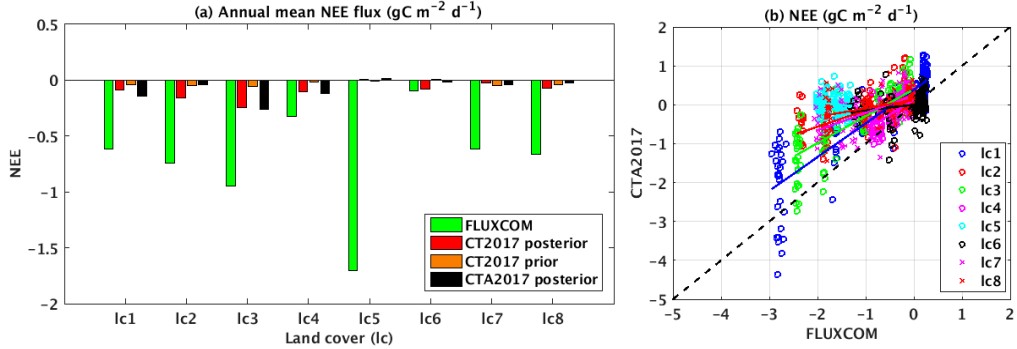

**Figure 9.** Annual mean NEE flux (**a**) and scatter plot for CTA2017 versus FLUXCOM on a monthly mean basis (**b**) during 2001–2013. Note that lc1-conifer forest, lc2-broadleaf forest, lc-3-mixed forest, lc4-grass/shrub, lc5-tropical forest, lc6-shrub/woods, lc7-fields/woods/savanna, and lc8-cropland.

**Table 5.** Annual mean NEE flux over different biomes in Asia.

| SS.N | Land Cover Type | Annual Mean NEE Flux (gC m$^{-2}$d$^{-1}$) | | | |
|:---:|:---:|:---:|:---:|:---:|:---:|
| | | CT2017 Priori | CTA2017 Posteriori | CT2017 Posteriori | FLUXCOM |
| **1** | Conifer forest | $-0.04 \pm 0.77$ | $-0.15 \pm 1.13$ | $-0.10 \pm 1.07$ | $-0.61 \pm 1.13$ |
| **2** | Broadleaf forest | $-0.05 \pm 0.27$ | $-0.04 \pm 0.50$ | $-0.16 \pm 0.60$ | $-0.74 \pm 0.80$ |

**Table 5.** *Cont.*

| SS.N | Land Cover Type | Annual Mean NEE Flux (gC m$^{-2}$d$^{-1}$) | | | |
|---|---|---|---|---|---|
| | | CT2017 Priori | CTA2017 Posteriori | CT2017 Posteriori | FLUXCOM |
| 3 | Mixed forest | −0.05 ± 0.50 | −0.26 ± 0.76 | −0.25 ± 0.81 | −0.95 ± 0.87 |
| 4 | Grass/shrub | −0.02 ± 0.28 | −0.12 ± 0.31 | −0.10 ± 0.31 | −0.32 ± 0.29 |
| 5 | Tropical forest | −0.01 ± 0.19 | −0.02 ± 0.23 | −0.01 ± 0.27 | −1.71 ± 0.22 |
| 6 | Scrub/woods | 0.006 ± 0.13 | −0.02 ± 0.28 | −0.08 ± 0.35 | −0.10 ± 0.41 |
| 7 | Fields/woods/savanna | −0.05 ± 0.39 | −0.04 ± 0.44 | −0.02 ± 0.51 | −0.62 ± 0.60 |
| 8 | Croplands | −0.04 ± 0.16 | −0.02 ± 0.30 | −0.07 ± 0.38 | −0.67 ± 0.60 |

### 3.3. Diurnal Averaged NEE Flux in Summer

This section is focused on diurnal averaged NEE estimates in the summer season. NEE flux is dominated by gross primary productivity (GPP) and ecosystem respiration (TER) during the daytime and nighttime, respectively. On the diurnal timescale, the variation of $CO_2$ due to uptake by plants through photosynthesis in sunlit hours is strongly modulated by turbulent transport through the planetary boundary layer (PBL), which also evolves throughout the day. This is the so-called rectifier effect, which helps to explain the annual mean north–south gradient of $CO_2$ [42]. Figure 10 depicts the diurnal cycle of NEE flux derived from CTA2017 and FLUXCOM during the summer season for 2001–2013. The overall result highlighted that a better agreement was noticeable during the carbon release period and the discrepancy was evident during the peak carbon uptake period; CTA2017 underestimated the carbon sink by a magnitude of around 3.2 gC m$^{-2}$ d$^{-1}$ against FLUXCOM. We also explored the spatial agreements between them. To do this, we presented 3 hourly NEE averages (3-hour intervals) for the whole year (2001–2013) for each grid cell of a given domain from June to August. Figure 11 demonstrates the summer diurnal mean estimates of NEE flux from CTA2017 (panels: a–d, i–l) and FLUXCOM (panels: e–h, m–p). We noted that the carbon sink was largely observed during 01:30–07:30 UTC, and carbon release was evident between 13:30 and 19:30 UTC. They exhibited similar features, but the magnitude of the sink and source between CTA2017 and FLUXCOM varied spatially and across individual PFTs. For instance, the inversion model detected a larger diurnal amplitude in boreal Eurasia and southeast China, with a maximum of −33 gC m$^{-2}$ d$^{-1}$, than FLUXCOM (with maximum of −19 gC m$^{-2}$ d$^{-1}$). The other noticeable difference was identified over India and Southeast Asia at 10:30 UTC, where the flux was estimated to be around −2.5 gC m$^{-2}$ d$^{-1}$ by CTA2017 and 9.5 gC m$^{-2}$ d$^{-1}$ by FLUXCOM. This comparison at a higher temporal resolution also highlighted the challenges in land cover classification when phenology has to be accounted for, as well as the impact of boundary layer dynamics. Despite not being able to explicitly quantify the drivers behind the disagreement, we suggest that there is an identified problem related to the FLUXCOM estimates over the tropics and the other main sources of uncertainties in inverse estimates of regional $CO_2$ surface–atmosphere fluxes are related to model errors in vertical transport within the PBL.

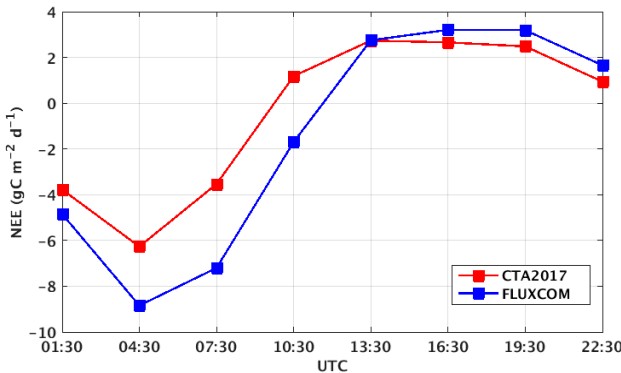

**Figure 10.** Summer diurnal NEE flux averaged over Asia domain during 2001–2013.

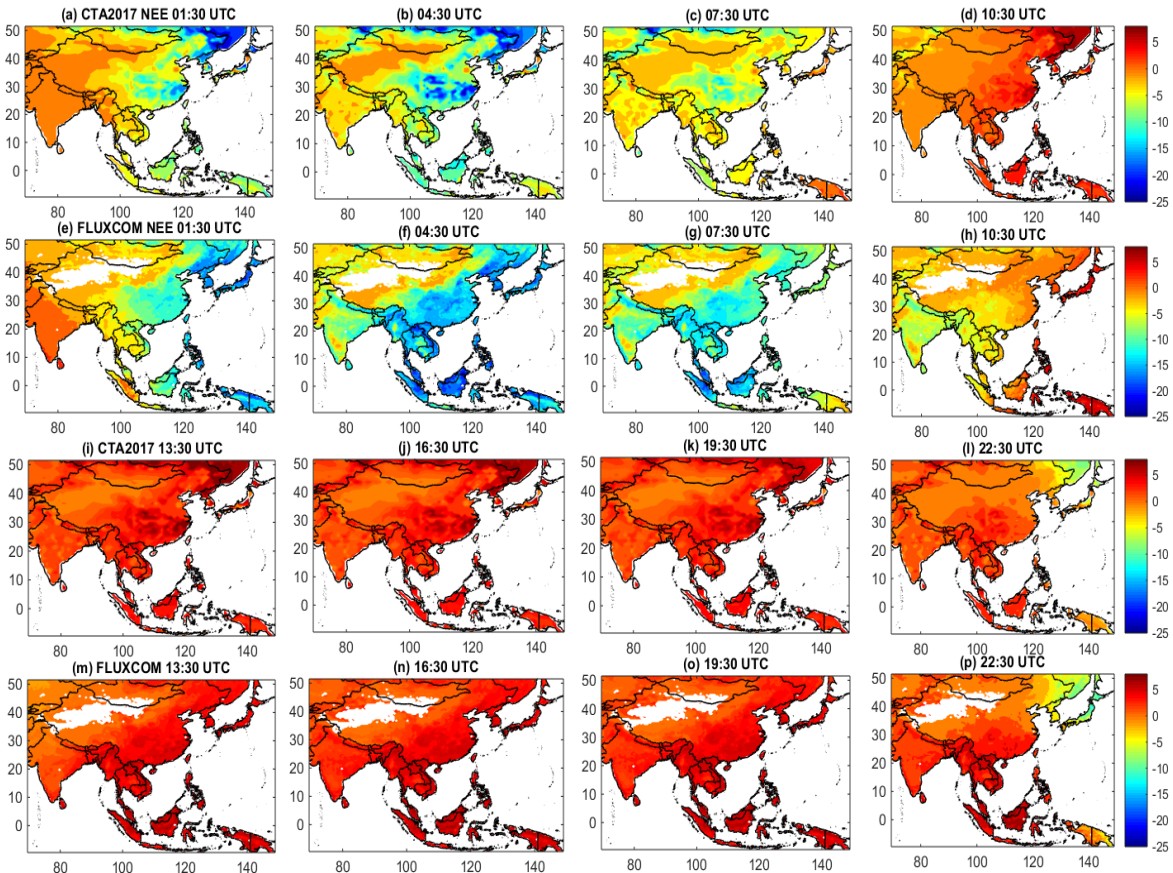

**Figure 11.** Summer diurnal mean of NEE flux (gCm$^{-2}$ d$^{-1}$) over Asia for 2001–2013. First and third rows are CTA2017 while second and fourth rows are FLUXCOM.

### 3.4. Spatio-Temporal Distribution of NEE Flux

Asia is one of the regions that exhibited a strong seasonal cycle of the terrestrial carbon uptake/sink (NEE), with the strongest carbon uptake in the middle (northeast temperate Eurasia) and high latitudes (northeast boreal Eurasia) of the Northern Hemispheric part of it. The low latitude (tropical Asia) region released $CO_2$ to the atmosphere during summer, whereas a reverse pattern was observed in winter, which is evidently observed in the simulated results as shown in Figure 12. Our result is consistent with previous findings indicating that northern temperate and high latitude ecosystems are strong carbon sinks [4,43,44] and tropical land regions are strong carbon sources [9].

Here, we presented annual and seasonally averaged posterior NEE flux (i.e., optimized) from CTA2017 and CT2017 during 2001–2013 in order to examine the impact of the nesting approach on the spatial pattern carbon sink and source strength. Figure 12 depicts the annual and seasonal means of NEE flux of CTA2017 (first columns), CT2017 (second columns), and their differences (CTA2017 minus CT2017 in the third columns). The overall pattern of the optimized NEE flux derived from those experiments was similar, but the overall magnitude depended on the ecosystem types and seasons. In general, the CTA2017 revealed a decrease of the annual mean carbon sink prevailing over the northeast region of the domain. This decrease of the carbon uptake was estimated about 0.20 gC m$^{-2}$ d$^{-1}$ predominantly across the ecosystem types of broadleaf forest, croplands, and semitundra over the Eurasia temperate region of the domain as compared to CT2017. On the other hand, in Eurasia boreal, the uptake was declined in mixed forest by about 0.40 gC m$^{-2}$ d$^{-1}$, and increased in confer forest by around −0.15 gC m$^{-2}$ d$^{-1}$. Interestingly, the feature of NEE flux shown from both experiments across the croplands in the northern Indian and Southeast tropical region is somewhat different. Particularly in northern India, CTA2017 exhibited less carbon release, a maximum of 0.34 gC m$^{-2}$ d$^{-1}$, than CT2017.

We further analyzed the detailed pattern in terms of season in order to distinguish the attributes of the seasons.

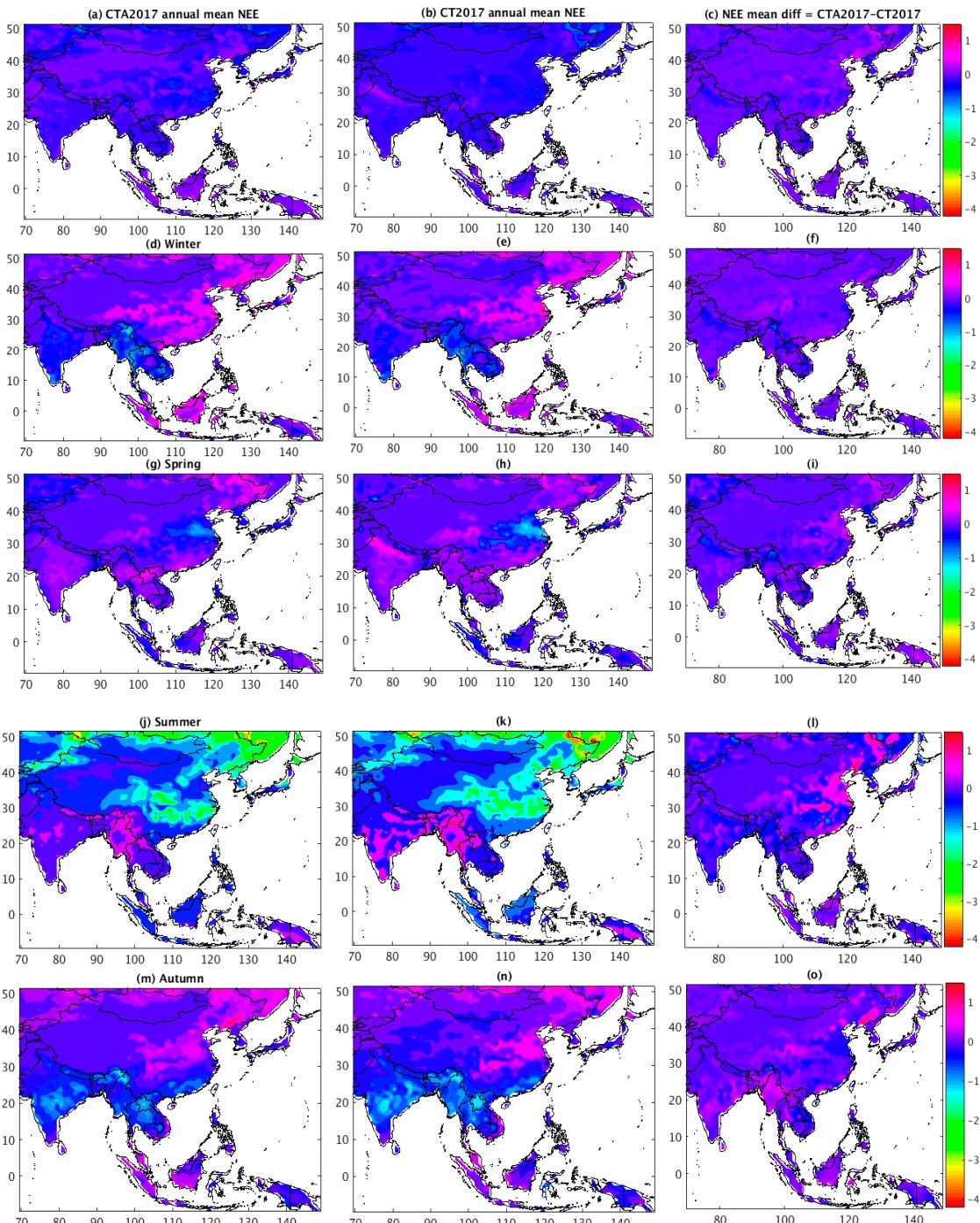

**Figure 12.** Annual and seasonal means of posterior NEE flux from CTA2017 (left panels) and CT2017 (middle panels), and CTA2017 minus CT2017 (right panels) over Asia. Unit NEE flux is gC m$^{-2}$ d$^{-1}$. Panels (**a**) CTA2017 annual mean, (**b**) CT2017 annual mean, (**c**) CTA2017-CT2017 annual mean; (**d**) CTA2017 Winter, (**e**) CT2017 Winter, (**f**) CTA2017-CT2017 Winter; (**g**) CTA2017 Spring, (**h**) CT2017 Spring, (**i**) CTA2017-CT2017 Spring; (**j**) CTA2017 Summer, (**k**) CT2017 Summer, (**l**) CTA2017-CT2017 Summer; (**m**) CTA2017 Autumn, (**n**) CT2017 Autumn, (**o**) CTA2017-CT2017 Autumn

In the winter season, CTA2017 revealed a higher carbon release, by a maximum of 0.41 gC m$^{-2}$ d$^{-1}$, in the south (mainly in the semitundra region) and southeastern China and northeastern Asia (mainly

in broadleaf forest region), than CT2017. In this season, the results from both inversions indicate that carbon uptake prevailed over India and Southeast Asia (includes Nepal, Bangladesh, Bhutan, Myanmar, Vietnam, and Cambodia). However, CTA2017 still exhibited low carbon uptake in the region. In spring, the strong carbon uptake was mostly confined to the croplands over eastern China, where CTA2017 estimates of carbon sink were lesser than CT2017, 0.48 gC m$^{-2}$ d$^{-1}$, while larger carbon release was evident in broadleaf forest in the northeast Asia, which is by 0.48 gC m$^{-2}$ d$^{-1}$. During summer, carbon release is predominant in the south and Southeast Asia, where the estimation made by the nested grid was lower again. In autumn, the carbon release estimate from CTA2017 was reduced, on average by 0.22 gC m$^{-2}$ d$^{-1}$, in the croplands in eastern China, but it was higher, by about 1.04 gC m$^{-2}$ d$^{-1}$, in the broadleaf forest in northeast Asia. In Section 3.2, we discussed the time series of monthly mean aggregated NEE flux for individual biomes. Some differences are observed in the strength of the carbon sink and source, which is linked to the transport model resolution differences, and differences in the number of observation data for constraining the flux in the region of interest.

As with the mean estimate, we also discerned spatial variations of the standard deviations (SD) of posterior NEE flux from both experiments. The SD value was computed from the monthly mean NEE values. As can be seen in the spatial distribution of SD of NEE (Figure 13), the overall pattern was similar, but the magnitude was somehow different at seasonal and annual mean time scales. The CTA2017 demonstrated that the annual estimate of SD was lower (by a maximum of −0.7 gC m$^{-2}$ d$^{-1}$, which is the difference of CTA2017 minus CT2017) over a large part of the area, except Southeast Asia. Over Southeast Asia, CTA2017 prevails with a larger SD, estimated to be about 0.50 gC m$^{-2}$ d$^{-1}$. This pattern was evident in winter, with a lower SD (~−0.50 gC m$^{-2}$ d$^{-1}$) across the region of interest, with the exception of Southeast Asia. In spring, CTA2017 revealed a larger SD (~0.50 gC m$^{-2}$ d$^{-1}$) in the northeast temperate broadleaf forest while during summer a strong decline was oppositely evidenced. The difference in SD of NEE between those experiments was found to be high in summer compared to other seasons. In this summer, CTA2017 showed a lower value of SD as high as −1.1 gC m$^{-2}$ d$^{-1}$, very clear in northeast broadleaf forest and sometimes in mixed forest. This difference is maintained in autumn, particularly in northeast broadleaf forest. Since the transport model resolution increases, it is expected to improve in capturing the detailed features of NEE flux, as well as increase the accuracy in quantifying the model uncertainty. Despite the seasonal variability, the larger SD seen by the CT2017 might be partially reflecting the uncertainties of the simulated flux. As the previous studies have indicated, the uncertainty is higher in summer than in winter over temperate and boreal land regions, given stronger convective transport and higher horizontal wind shear in the summer months [13].

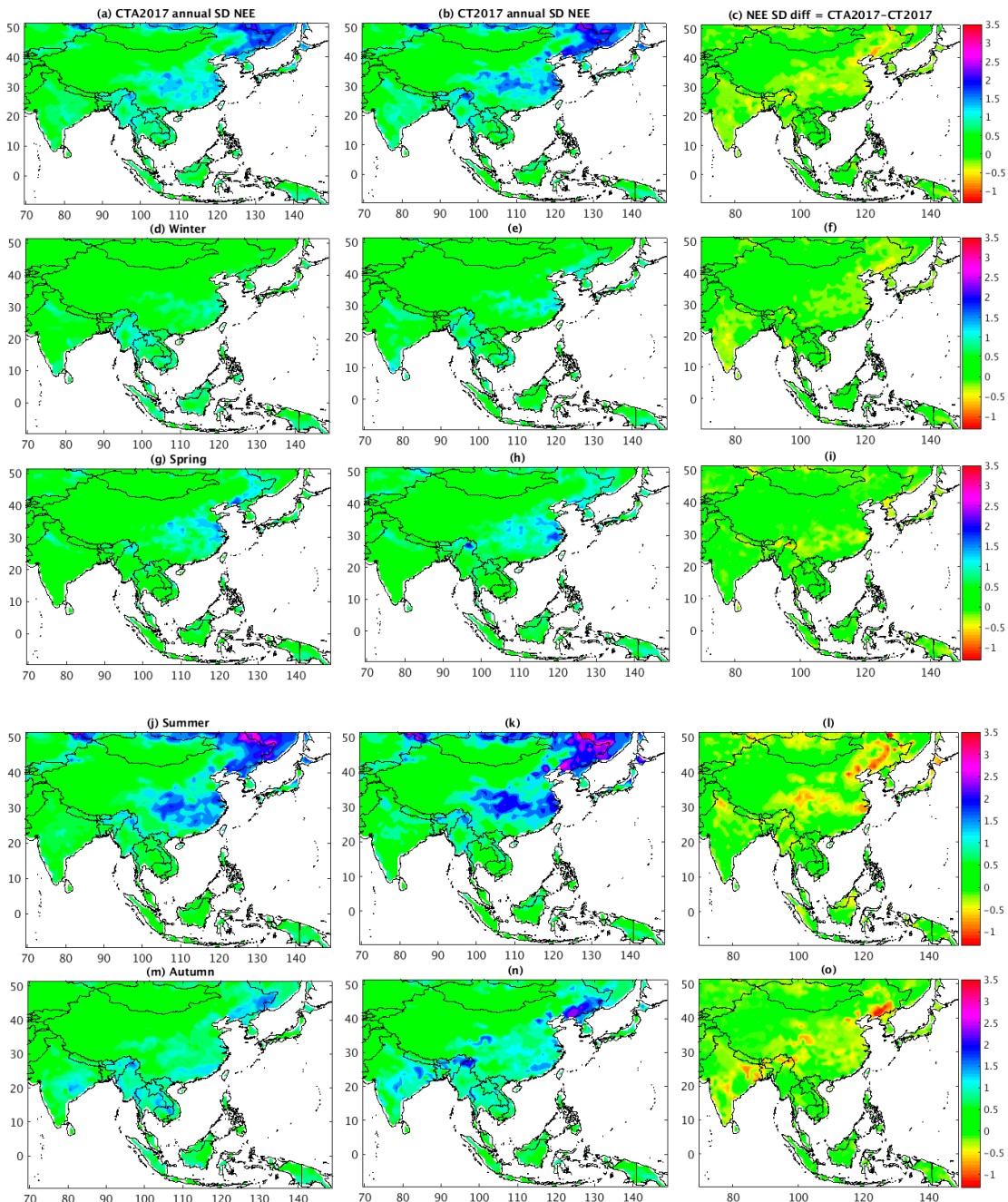

**Figure 13.** Similar with Figure 12 but for standard deviations (SD) of posterior NEE flux. Panels (**a**) CTA2017 annual SD, (**b**) CT2017 annual SD, (**c**) CTA2017-CT2017 annual SD; (**d**) CTA2017 Winter, (**e**) CT2017 Winter, (**f**) CTA2017-CT2017 Winter; (**g**) CTA2017 Spring, (**h**) CT2017 Spring, (**i**) CTA2017-CT2017 Spring; (**j**) CTA2017 Summer, (**k**) CT2017 Summer, (**l**) CTA2017-CT2017 Summer; (**m**) CTA2017 Autumn, (**n**) CT2017 Autumn, (**o**) CTA2017-CT2017 Autumn.

## 4. Conclusions

We analyzed the consistency and discrepancy of the optimized NEE flux over Asia from CTA2017 through a comparison with FLUXCOM and other global inversions results to better understand the uncertainties of NEE in the region during the period of 2001–2013. While comparing CTA2017 with the global inversions, such as CAMS, MACC, Jena CarboScope, and FLUXCOM, the carbon uptake obtained from CTA2017 was smaller than CAMS, MACC, Jena, and FLUXCOM. The annual average of CTA2017, CAMS, MACC, Jena CarboScope, and FLUXCOM estimates were −0.32, −0.42, −0.58,

−0.81, and −5.86 PgC yr$^{-1}$, respectively. The use of different prior fluxes between the CTA2017 and global inversions could be responsible for the discrepancy in the optimized fluxes. We found that the influence of prior flux in MACC and CAMS is somehow strong as compared to CTA2017 during the peak carbon uptake season. Some differences were also observed between MACC and CAMS, which might be attributed to the differences in the number of observation data used for assimilation, since they have the same transport model and inverse set-up. The inversion result revealed a large discrepancy with FLUXCOM, and this was mainly due to the FLUXCOM estimates of NEE over the tropical forest. Furthermore, we showed the level of agreement for the temporal aggregation of NEE for individual biomes between CTA2017 and FLUXCOM at a monthly resolution. The overall results demonstrated good consistency of the time series of monthly NEE over conifer forest, broadleaf forest, mixed forest, grass/shrub, and fields/woods/savanna; poor agreement in croplands and scrub/woods; and the worse agreement in tropical forest land. The key problems are underrepresentation of FLUXCOM NEE estimates by limited eddy covariance flux measurements, the role of $CO_2$ emissions from land use change not accounted for by FLUXCOM, sparseness of surface observations of $CO_2$ concentrations used by the assimilation systems, and land cover inconsistency between top-down and bottom-up systems.

The level of agreement on the diurnal cycle of aggregated NEE averaged over the domain between CTA2017 and FLUXCOM was examined. The result revealed that better agreement was found during the carbon release period than the carbon uptake period. Summer diurnal averaged NEE flux spatial distributions between CTA2017 and FLUXCOM were also compared. To do this, we provided 3 hourly NEE averages (3-hour intervals) for the whole year (2001–2013) for each grid cell of a given domain. The inversion model detected a larger diurnal amplitude in boreal Eurasia and Southeast China, with a maximum of −33 gC m$^{-2}$ d$^{-1}$, than FLUXCOM (with a maximum of −19 gC m$^{-2}$ d$^{-1}$). The other noticeable difference was evident over tropical Asia at 10:30 UTC. Nevertheless, we cannot explicitly list all the drivers behind the discrepancy, we suggest that there is a known problem related to the FLUXCOM estimates over tropics, and the other key sources of uncertainties in inverse estimates of regional $CO_2$ surface–atmosphere fluxes is related to model errors in vertical transport within the PBL.

We examined the impact of nesting on the spatial pattern of the strength of the carbon sink and source by comparing the CTA2017 and CT2017 optimized NEE fluxes in terms of seasonal and annual time scales over the region of interest. Before that, we showed that both inversion optimized NEE fluxes were dissimilar with the prior NEE flux in magnitude, depending on seasons and ecosystem types, which implies they captured additional biospheric surface flux signal, e.g., this difference was substantial in the conifer forest region. Within the experiments, we noted some differences in the magnitude of the carbon sink and source emerge both spatially and temporally. For the annual average, CTA2017 indicated a carbon uptake reduction by a magnitude of ~0.20 gC m$^{-2}$ d$^{-1}$ largely across broadleaf, cropland, and semitundra ecosystems over the Eurasia temperature region. Furthermore, all magnitudes of the carbon sink and source varied within the experiments. For example, during summer, carbon release prevailed in tropical Asia, where the estimation shown by CTA2017 was lower than CT2017. The annual aggregated NEE estimates of CTA2017 and CT2017 were −0.32 and −0.58 PgC yr$^{-1}$, respectively. Both models unveiled IAV of NEE flux, but their magnitudes were different and this led to differences in the annual aggregated results. Differences in the estimated IAV of NEE in response to ENSO may stem from the differences in transport model resolutions.

This finding suggests that a detailed investigation on the FLUXCOM and inverse estimates is most likely required not only in temperate regions of Asia (as indicated by previous studies) but also in the tropics. In addition, this study also recommends further investigation on how the updates made in CarbonTracker affect the interannual variability of the aggregate and spatial pattern of NEE flux in response to the ENSO effect over Asia.

**Author Contributions:** Writing—original draft preparation, S.T.K.; writing—review and editing, L.D.L., T.-Y.G., S.L., Y.-S.O., and Y.-H.B. All authors have read and agreed to the published version of the manuscript.

**Funding:** This work was funded by the Korea Meteorological Administration Research and Development Program "Development and Assessment of IPCC AR6 Climate Change Scenario (1365003000)".

**Acknowledgments:** The authors are very grateful for Jae-Sang Rhee for his valuable contribution for the simulated surface $CO_2$ flux data from CarbonTracker-Asia model. We also strongly acknowledge for those who provided to access the data from Jena CarbonScope, CAMS, MACC, FLUXCOM, and MODIS.

**Conflicts of Interest:** The authors declare no conflict of interest.

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
