# Peer review of "Comparison of Regional Simulation of Biospheric CO2 Flux from the Updated Version of CarbonTracker Asia with FLUXCOM and Other Inversions over Asia"

_remotesensing, doi:10.3390/rs12010145_

Round 1

Reviewer 1 Report

This study “Comparison of regional simulation of biospheric CO2 flux from the updated version of CarbonTracker Asia with FLUXCOM and other inversions over Asia” used NEE from CarbonTracker and other inversion models and FLUXCOM to show its spatial and temporal variations over Asia. It is good to know how the CarbonTracker, inversion models and FLUXCOM perform in the Asia, though there are already several studies exploring the performance of NEE from different sources (CarbonTracker, inversion models, etc.). The results are demonstrated clearly. However, I have some comments regarding to the current manuscript.

This study used FLUXCOM NEE as kind of benchmark to evaluate NEE from CarbonTracker and other inversion models. However, FLUXCOM NEE did not consider the CO2 fertilization and greening impacts on terrestrial ecosystem carbon uptake. It is good to use FLUXCOM NEE to show the spatial patterns, however, the temporal variations of NEE from FLUXCOM may not be the truth. Thus, the difference between FLUXCOM NEE and NEE from inversion models may be due to the uncertainties in FLUXCOM NEE, and the inversion models may have the right NEE values (Ln. 292-293). There are several flux tower sites in Asia, why do the authors not use the NEE from flux tower measurements for comparisons, since flux tower measured NEE could be served as ground truth? The first objective of this study is to explore the impact of nesting approach on the optimized NEE flux over Asia at seasonal and annual timescales (Ln. 78-79), however, I could not get this point through reading the manuscript. How to define or choose the optimized NEE flux? Why the NEE from different models are different from each other? Because of different inputs? Please provide analysis on these with appropriate references. 177-184, why there is an extra Method session? The Section 2 is Materials and Methods. Table 3, what does the color represent for?

Reviewer 2 Report

Comments for Manuscript: Remote SensingManuscript number: 647874

Title: Comparison of regional simulation of biospheric CO2 flux from the updated version of CarbonTracker Asia with FLUXCOM and other inversions over Asia  

This article makes an effort to assess the impact of nesting approach on the optimized NEE flux derived from CarbonTracker Asia and compare its NEE estimates with FLUXCOM and the global inversion models from CAMS, MACC and Jena CarboScope in Asia during the period of 2001-2013. It is an ambitious study on substantial amounts of data, and the authors claim that this assessment is more accurate than the many previous articles on this subject, which may be true. However, I consider also this study to comprise several sources of uncertainty that are not at all treated or only loosely discussed. Thus my assessment is that the article cannot be accepted in its current shape but needs to be improved.

My detailed comments are:

In general, there is a need for some improvements of the language and checks and fixes of details. For example, L46-47 the sentence “Previous findings revealed large remained uncertainties still in the estimates of net ecosystem 46 exchange of CO2 (NEE) in Asia.” Some references are outdated, not typical. It is recommended to quote the latest published results. In the Introduction, there is not sufficient evidence to support this research background. For example, why is it important to study the updated version of CTA? How to update CTA? Are there any scientific implications? A poor order and quality of Figures and section headings occurs in the main text. In addition, wrong titles of Figures and Tables are used in the manuscript. Please check it throughout the paper. Importantly, there is no introduction and assessment of the updated CTA in the Methods sections, which may be quite necessary. How to update it? Compared with previous versions, its advantage? In Results section, the compared results should be showed firstly, that is, the accuracy assessment and validation of the updated CTA compared with previous studies. Is it possible to obtain field measurements? Then, to analyze and discuss the spatial and temporal distributions. Figures (e.g. 4 and 5) should be placed behind the relevant statements in the main text. Many different sources of information are mentioned in the article, but it is not altogether clear what kind of information were retrieved from the different sources and how the information was used. For example, where do the data come from? Who provide the data and for which purpose? Was a quality and compatibility check done? Organization is poor. Methodology, Results and Discussion sections should be reorganized to clarify the research. There is some specific comments highlighted in the manuscript.

Reviewer 3 Report

This study focuses on data analysis of large uncertainties in the estimation of net ecosystem exchange of CO2 in Asia from the updated version of CarbonTracker (CT2017) (with and without nested-grid Asia), through comparison with FLUXCOM and other global inversions results. The results are interesting for data users, future analyses and modeling studies. However, significant disadvantages are noticeable. The structure of the work is not well thought out and lacks important details. The analysis does not fully utilize the potential of the considered datasets. The quality of the figures is not enough for a quick understanding of the topic under discussion. Captions to them are not detailed enough.

The major concerns of this study are:

Multiple notations of the CarbonTracker (CTA-CT2017, CTA, CT, CTA2017, CT2017, CTA-exp, CT-exp) products are very confusing (especially for Fig. 1-2). Notations used at the website (http://www.nimr.go.kr/2/carbontracker/different.html) make the situation even more misleading. Please be consistent and clearly describe the CarbonTracker products. Effect of additional observations (Ryori, Yonagunijima, and Minamitorishima and also aircraft observation data from CONTRAIL) in the CT-exp inversion was not reviewed. It is not described how the CONTRAIL aircraft observation data was used in the CT-exp inversion. It is not clear where the FLUXCOM sites are located (please plot sites on a map) and how the location of the stations affects the measurement results. The FLUXCOM data have a large number of missing points. Please explain why and how it could affect the analysis (for example, for the case of the domain aggregated monthly mean NEE flux comparison). It is not clear how CAMS, MACC, and Jena CarboScope products are correlated to FLUXCOM fluxes. A more detailed intercomparison between CAMS, MACC, Jena CarboScope vs FLUXCOM would be of particular interest for the scientific community. Please add more datasets into the figures 6-11 analysis.

Other specific comments are as follows:

Fig.1-2: poseriori -> posteriori Fig.2: Please use the same color scale for comparison. Fig.3: Please use the same color scale for comparison. L236: “The nested-grid Asia” is confusing. Please revise. L249: What does it mean “P-value”? Please clarify. Fig.4a-c: Please use the same color scale for comparison. Fig.4d: Color bar scale is too raw. Please update. L259: Please provide more details regarding how spatial Pearson’s correlation coefficient was computed. How long are the datasets? What is a purpose for the comparison of old and new versions of CarbonTracker (CT2015, CT2016, CT2017, CTA-2016, CTA-2017)? Fig.7: The caption for panel “c” is missing. L282-285: “… which might be attributed to largely due to the set-up of different prior flux. In addition to prior flux, in smaller spatial scale, the model resolution, transport model differences, and the meteorological drivers could be the possible reasons for the occurrence of differences of the flux estimates.” This statement is quite trivial. Adding more details will make this work more meaningful for the scientific community. References to Fig. 7a-b and Fig. 8 are missing. Fig.10: It is not clear why the CTA fluxes have such a large spread. For example, the CTA lc1 is in the frame of (-4.5…0.5) for FLUXCOM lc1 ≈-2.8. Fig.12 is very confusing due to different color scales for each panel. Please use one.

Round 2

Reviewer 1 Report

I think the authors addressed all of my comments, and the current manuscript could be accepted in Remote Sensing

Author Response

All improvements made in the current version are highlighted by red color.

Reviewer 3 Report

This study focuses on data analysis of large uncertainties in the estimation of net ecosystem exchange of CO2 in Asia from the updated version of CarbonTracker (CT2017) (with and without nested-grid Asia), through comparison with FLUXCOM and other global inversions results. The results are interesting for data users, future analyses and modeling studies.

In the revised version of the article, the previously mentioned comments were clarified and corrected. I recommend the work for publication after clarifying one issue:

It is unclear how FLUXCOM obtains fluxes in Southeast Asia between 0 and 20 degrees north latitude, where there are no stations for observations. Please explain.
